

# Rain-on-wet-soil compound floods in lowlands: the combined effect of large rain events and shallow groundwater on discharge peaks in a changing climate

Claudia C. Brauer[1], Ruben O. Imhoff[2], and Remko Uijlenhoet[3]

[1]Hydrology and Environmental Hydraulics Group, Wageningen University, Wageningen, The Netherlands
[2]Department of Operational Water Management & Early Warning, Unit of Inland Water Systems, Deltares, Delft, The Netherlands
[3]Department of Water Management, Faculty of Civil Engineering and Geosciences, Delft University of Technology, Delft, The Netherlands

**Correspondence:** Claudia C. Brauer (claudia.brauer@wur.nl)

**Abstract.** In lowland catchments, the severity of pluvial floods is determined by both the magnitude of rainfall events and the initial catchment wetness. The aim of this study was to determine the importance of initial wetness on flood peaks in lowland catchments and to examine if and how this affects the magnitude and timing of floods in the future. We used the rainfall-runoff model WALRUS to investigate the relation between initial groundwater depth (48 hours before the peak), effective rainfall sum (over the 48 hours before the peak) and the resulting peak discharge and peak volume in 12 lowland catchments, for 109 years of forcing in the current climate and four climate scenarios for both 2050 and 2085. We found that this relation is strong in these catchments, with a stronger dependence on initial groundwater depth in flatter catchments. When climate changes, less precipitation and more evapotranspiration are projected in summer, resulting in deeper groundwater in summer and autumn, reducing flood frequency and magnitude. More rain in autumn, winter and spring will lead to more severe floods in winter and spring only. Averaged over all catchments, scenarios and seasons, the effective rainfall sum is projected to increase with 1.5 % in 2050 and 5.6 % in 2085, while the initial groundwater depth increases with 0.7 % in 2050 and 0.3 % in 2085. This combination leads to more frequent and severe floods, with 1 % more floods and 3 % larger peak volumes in 2050 and 9 % more floods and 21 % larger peak volumes in 2085. Without the mitigating effect of the deeper initial groundwater tables, the higher rainfall sums would have led to more frequent and more severe floods in the future.

## 1 Introduction

The effect of antecedent wetness, in particular soil moisture, on flood risk has been investigated in many areas around the world (e.g. Borga et al., 2019; Garg and Mishra, 2019). The relation between catchment wetness and discharge response is especially strong in lowland catchments, which we define as areas with shallow groundwater and limited topography, where the shallow groundwater, where groundwater is shallow and therefore determines the flowpaths of rain water towards the surface water. When groundwater exceeds certain thresholds, drainpipes are activated (Tiemeyer et al., 2007; Van der Velde et al., 2010; King et al., 2014; Hansen et al., 2019; Rathore et al., 2024), macropores fill up (Christiansen et al., 2004) and ponding and saturation





excess overland flow occur (Deasy et al., 2009; Appels et al., 2011). Understanding the joint effect of wet initial conditions and large rainfall events on flooding in such lowland catchments is not only relevant for water management under current climatological conditions, but also for future conditions, especially given their high population density and the importance of lowland (delta) catchments for food production. However, changes in precipitation statistics cannot be translated to floods directly. The seasonal differences in meteorological and hydrological processes, including dry spells, evapotranspiration and the catchment's drainage rate, determine the antecedent wetness conditions, which in turn determine part of the hydrologic response to rain (Wasko and Sharma, 2017; Blöschl et al., 2019; Cao et al., 2019; Brunner et al., 2021).

There is an abundance of studies in which long series of river discharge observations were analysed to determine past changes in the type of flood, the flood magnitude and the timing. In the following three paragraphs, we discuss some of these studies, focussing mostly on studies in western Europe in general and lowland areas in particular. Tarasova et al. (2020b) developed a method to classify floods in Germany, expanding the flood typology by Merz and Blöschl (2003) with additional attention to catchment wetness. They tested the method on 392 catchments in Germany, including several in lowland areas. They found that in the northwestern lowlands in Germany most floods occur under wet conditions after a significantly long period of rain (especially for less severe events, as they found in their follow-up paper; Tarasova et al., 2020a), whereas the number of floods with wet antecedent conditions and dry antecedent conditions was almost equal. Tarasova et al. (2023) found that, in western European countries, floods caused by rain events on wet soils have become more frequent over the last decades. Fischer et al. (2019) distinguished rain-induced floods in a different manner, not by initial wetness, but by the ratio between flood volume and flood peak, as an indicator for the flashiness of the event. They found that the frequency of floods caused by heavy rain in western Germany has increased over time, but found no change for floods with a moderate or low intensity. Berghuijs et al. (2019) complemented a discharge time series analysis for Europe with a simple soil moisture model and found that in the region around the Netherlands, most floods are caused by soil moisture excess. It should however be noted that lowland-specific processes related to the tight connection between saturated and unsaturated zone were not included in their event water balance model to simulate soil moisture. Interactions between antecedent wetness and flood magnitude have also been observed by Liu et al. (2022) and Jiang et al. (2022), who analysed a large number of past flood events around the world and attributed these to different drivers (e.g. snow, heavy rain, wet initial conditions).

In terms of flood magnitude, the results vary. Bertola et al. (2020) found that in northwestern Europe flood severity has increased. In small catchments severe floods were found to increase more than moderate floods, while in larger catchments the opposite is the case. Blöschl (2022) found that floods increase in very small catchments as a result of increasing precipitation intensity, but that this effect is smaller in larger catchments, where the seasonality of soil moisture and snow are more important.

The timing of floods was found to shift in some areas. Beurton and Thieken (2009) compared the timing of discharge peaks in Germany in different historical periods and found that discharge stations in northwest Germany (including lowlands) experienced mostly winter floods throughout the analysed period, while northern catchments (with limited topography) had a larger contribution of summer floods before 1970 than after. Blöschl et al. (2017) used time series from 4262 European discharge stations from 1960 to 2010 to determine if the timing of floods has shifted and found that near the North Sea the maximum annual floods usually occur in winter but that there occurrence was delayed by two months more recently, likely



due to the changed climate. They attribute this time shift to changes in timing of the precipitation rather than changing initial conditions. In contrast, Villarini (2016) did not find evidence for changes in seasonality over a large number of streamgauges in the US.

Although a few of the streamgauges included in the studies comparing European rivers (e.g. Blöschl et al., 2017; Berghuijs et al., 2019) are located in lowland areas, these gauges are mostly situated in large rivers (Hall et al., 2015). The changes observed at those stations therefore do not reflect the changes in the smaller lowland catchments surrounding the streamgauge, but reflect the change in the whole catchment upstream, including the mountainous headwaters. Therefore, the lowland hydrological processes are not examined. In addition, stations with high anthropogenic impact are excluded from many studies,

which rules out most lowland areas since their population density is high and land and water are managed intensively. Discharge observations in smaller lowland rivers are often hampered by human interference (e.g. changing settings of the weir at which discharge is measured or redirection of water to and from channels) and measurement challenges (drowning, backwater effects). In addition, water management (and consequently monitoring) in lowland areas is mostly centered around controlling water levels rather than discharges. Since in these managed channels relations between water level and discharge vary, water

level observations cannot be translated to discharges directly. Therefore, historical changes in flood type, magnitude and timing in (small) lowland catchments remain largely unknown.

To overcome the lack of historical data and to be able to look ahead, hydrologic researchers and water managers often use climate scenarios in combination with hydrological models. Climate projections such as EURO-CORDEX (Jacob et al., 2014), CMIP (Taylor et al., 2012; O'Neill et al., 2016) and, more specific, the climate scenarios for the Netherlands by KNMI (Royal

Netherlands Meteorological Institute,  KNMI, 2015; Van Dorland et al., 2023) are available for this purpose. Such scenarios are based on climate model runs for different socio-economic pathways and associated greenhouse gas emissions. Climate model output is used directly (or in downscaled form) in some hydrologic applications, while in other applications, these provide the basis for transformations of observed time series.

However, studies using climate scenarios for discharge simulations in lowlands are scarce because most rainfall-runoff

models are not applicable there. Discharge simulation in lowland areas is not trivial, since many feedbacks occur: groundwater and surface water are tightly coupled (surface water level management limits drainage or can even cause infiltration) and the saturated and unsaturated zone are in fact one continuous system (shallow groundwater causes capillary rise). In addition, the outline of lowland catchments is often not clearly defined, with groundwater flowing across the catchment boundaries (particularly seepage) and surface water supply. To be able to simulate discharge in such catchments for long time series, one

needs a model that is both computationally efficient and which includes these relevant aspects. The conceptual rainfall-runoff model WALRUS (Brauer et al., 2014a) meets these requirements.

The aim of this study is to determine the importance of initial wetness on flood peaks in lowland catchments and to examine if and how this affects the magnitude and timing of floods in the future. We used a 109-year hourly forcing time series and rainfall-runoff model WALRUS (Brauer et al., 2014a) to analyse the relation between effective rainfall sum, initial groundwater

depth and discharge peaks for 12 lowland catchments in the Netherlands and just across the border in Germany and Belgium. We use simulated groundwater depth and discharge rather than observations because observed time series are too short for





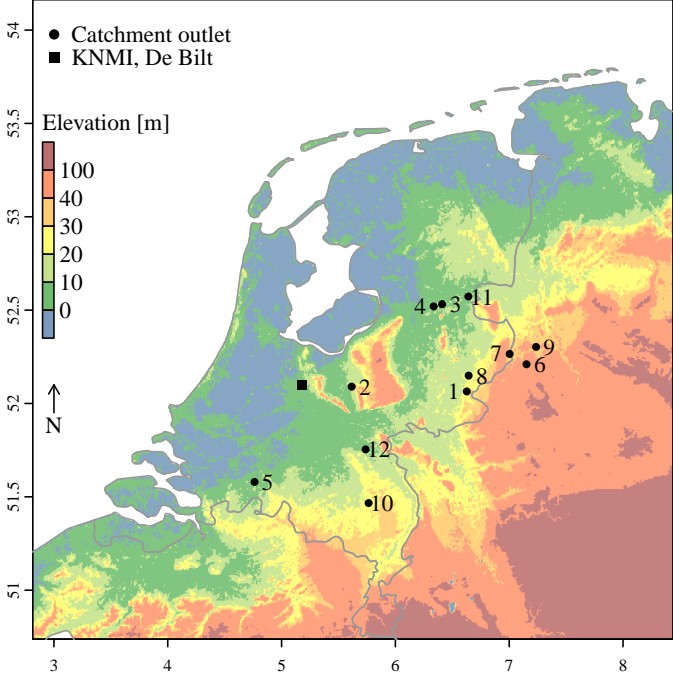

**Figure 1.** Locations of the 12 catchments in the Netherlands and the meteorological observations (De Bilt). See Tab. 1 for names and characteristics of the catchments.

robust statistical analyses. We examine differences between catchments and evaluate how climate change affects the conditions leading to moderately high and extremely high discharges. The combination of a dedicated rainfall-runoff model and long time series with hourly resolution allows us to obtain a detailed look at the interplay between initial conditions and flood severity

for a range of lowland catchments, while focusing on floods with high return periods.

## 2 Methods

### 2.1 Catchments

We used 12 catchments in the east and south of the Netherlands or just across the border in Germany or Belgium (Fig. 1) which together represent a broad range of conditions found in freely draining lowland areas in delta landscapes. Catchment sizes

range from 6.5 to 2821 km$^2$. All catchments are freely draining and have mostly sandy soils, but in two catchments upward seepage occurs and in three catchments surface water is supplied (Tab. 1). Land use is predominantly agricultural (in particular grass and maize), with small patches of forest or urban areas. The catchments vary in slope and aquifer properties, leading to differences in discharge response times and seasonality.



**Table 1.** Catchment-specific WALRUS input: seepage ($f_{XG}$), surface water supply ($f_{XS}$), model parameters and catchment characteristics. Note that surface water is only supplied between 1 Apr. and 30 Sep. The catchments are ordered from high to low discharge threshold leading to on average 10 peaks per year (Fig. 3).

| No. | Catchment | Additional forcing | | Model parameters | | | | | | | Catchment characteristics | |
|---|---|---|---|---|---|---|---|---|---|---|---|---|
| | | $f_{XG}$ | $f_{XS}$ | $c_W$ | $c_G$ | $c_Q$ | $c_V$ | $c_S$ | $c_D$ | $a_S$ | Soil type | Size |
| | | [mm d$^{-1}$] | [mm d$^{-1}$] | [mm] | [$10^6$ mm h] | [h] | [h] | [mm h$^{-1}$] | [m] | [$-$] | | [km$^2$] |
| 1 | Hupsel Brook | 0 | 0 | 356 | 5 | 3 | 0.2 | n.a.[1] | 1.5 | 0.01 | Hupsel[1] | 6.5 |
| 2 | Steinfurter Aa | 0 | 0 | 275 | 8.4 | 1.7 | 9 | 1.2 | 1.85 | 0.01 | sand | 205 |
| 3 | Luntersebeek | 0 | 0 | 200 | 30 | 20 | 10 | 2 | 1.5 | 0.01 | sand | 39 |
| 4 | Aa of Weerijs | 0 | 0 | 245 | 25 | 20 | 10 | 4 | 1.8 | 0.01 | sand | 287 |
| 5 | Ommerkanaal | 0 | 0.50 | 330 | 0.5 | 44 | 27 | 4.9 | 1.45 | 0.01 | loamy sand | 171 |
| 6 | Vechte A | 0 | 0 | 261 | 9.4 | 49 | 15 | 7.8 | 1.7 | 0.01 | loamy sand | 183 |
| 7 | Bakelse Aa | 0.26 | 0.35 | 299 | 8 | 43 | 10 | 4 | 1.8 | 0.015 | sand | 90 |
| 8 | Dinkel | 0 | 0 | 395 | 15 | 33 | 10 | 4 | 2.4 | 0.01 | sand | 643 |
| 9 | Vecht | 0 | 0 | 394 | 74 | 88 | 31 | 10.7 | 2.2 | 0.01 | loamy sand | 2821 |
| 10 | Lage Raam | 0.20 | 0 | 350 | 4.5 | 50 | 30 | 1.3 | 1.91 | 0.015 | sand | 161 |
| 11 | Grote Waterleiding | 0 | 0.22 | 240 | 20 | 35 | 10 | 3 | 2.2 | 0.01 | loamy sand | 40 |
| 12 | Radewijkerbeek | 0 | 0 | 353 | 50 | 92 | 29 | 4 | 2.5 | 0.01 | sand | 106 |

[1] For the Hupsel Brook (Dutch name: Hupselse Beek) catchment, the stage-discharge relation was taken from the flume at the outlet (Brauer et al., 2014b) and the soil physical parameters were derived from local soil moisture observations (Brauer et al., 2014a).

[2] References for calibration: Brauer et al. (2014b, no. 1), Loos (2015, no. 2, 5, 6, 9), Imhoff et al. (2022, no. 3), Heuvelink et al. (2020, no. 11). Parameters for no. 10 were provided by the local water authority. Catchments 4, 7, 8 and 12 were (re)calibrated for this study.

## 2.2 Forcing

Hourly precipitation ($P$) and potential evapotranspiration ($ET_{pot}$, computed with the method of Makkink (1957)), were measured in De Bilt from 1906 to 2014 (the location of the Royal Netherlands Meteorological Institute, KNMI, Fig. 1). These series have been corrected for changes in the measurement set-up and detrended to make them representative for the climate of 2014 by Beersma et al. (2015). For the detrending procedure, the ratio between average precipitation measured over several years in a certain season was compared to 2014. This was done for each year in the past, resulting in correction factors for each season and each year. Then the observed time series was transformed linearly with these correction factors. The same method was used for temperature and global radiation, which were used to compute potential evapotranspiration.

Beersma et al. (2015) then transformed these detrended time series to the climate of 2050 and 2085 for four KNMI'14 climate scenarios (KNMI, 2015; Lenderink et al., 2014, based on Bakker and Bessembinder (2012)). The climate scenarios are combinations of changes in global temperature (moderate [G] or warm [W]) and circulation pattern (little change [L] or much change [H]). The scenarios were made using model runs from the Climate Model Intercomparison Project (CMIP5 Taylor et al., 2012), which included the climate model EC-Earth (Hazeleger et al., 2012), downscaled with RACMO2 (Van Meijgaard et al., 2008).





**Table 2.** Percentage increase/decrease in annual/seasonal sum of precipitation and potential evapotranspiration, and in the number of winter/summer days with daily precipitation sum above 10/20 mm, according to Van den Hurk et al. (2014). Note that these are not the result of a fixed multiplication factor, but average changes.

| | 2050 | | | | 2085 | | | |
|---|---|---|---|---|---|---|---|---|
| | GL | GH | WL | WH | GL | GH | WL | WH |
| $\Sigma P_{\text{year}}$ | 4 | 2.5 | 5.5 | 5 | 5 | 5 | 6 | 7 |
| $\Sigma P_{\text{winter}}$ | 3 | 8 | 8 | 17 | 4.5 | 12 | 11 | 30 |
| $\Sigma P_{\text{spring}}$ | 4.5 | 2.3 | 11 | 9 | 8 | 7.5 | 13 | 12 |
| $\Sigma P_{\text{summer}}$ | 1.2 | -8 | 1.4 | -13 | 1.0 | -8 | -4.5 | -23 |
| $\Sigma P_{\text{autumn}}$ | 7 | 8 | 3 | 7.5 | 7.5 | 9 | 5.5 | 12 |
| $\Sigma ET_{\text{pot,year}}$ | 3 | 5 | 4 | 7 | 2.5 | 5.5 | 6 | 10 |
| #winter days $P > 10$ | 9.5 | 19 | 20 | 35 | 14 | 24 | 30 | 60 |
| #summer days $P > 20$ | 4.5 | -4.5 | 6 | -8.5 | 5 | -3.5 | 2.5 | -15 |
| | : | : | : | : | : | : | : | : |
| (range) | 18 | 10 | 30 | 14 | 23 | 14 | 35 | 14 |

For the transformation, Beersma et al. (2015) first computed monthly values for changes in precipitation, temperature and global radiation based on the climate scenarios. Then, hourly values in the reference series were aggregated to daily values. For precipitation, the change in number of wet days and amount of rain on wet days was determined. Then, wet days were added to or removed from the reference time series, while preserving the probability distribution of rainfall amounts and accounting for timing (not interrupting consecutive rainy days). Next, a power-law transformation was applied to the wet days to increase or decrease the amount of rain on a certain day. Temperature was transformed using linear quantile scaling and global radiation using a linear transformation. The transformed reference evapotranspiration was then computed with the method of Makkink (1957). Then the daily values were disaggregated to hourly values using the original distribution over the hours. Since the series for the future climates are transformations of the original ones, specific events occur in all datasets, which allows direct comparison of their conditions and characteristics.

This resulted in nine time series of 109 years of hourly values for this study: one representative for the current climate and four for the different climate scenarios considered for both 2050 and 2085. In all scenarios annual precipitation increases and the number of days with high precipitation increases strongly in winter (Tab. 2; Van den Hurk et al., 2014). For days with high precipitation in summer, a range is given since the spatial variation is large for these small-scale events. In the GH and WH scenarios more dry summers occur.

We used one forcing time series for all catchments because the detrended and projected time series were only available for De Bilt. This was warranted since the variation in climate within the Netherlands is limited (KNMI, 2024). The distance between the furthest catchment and De Bilt is about 150 km.





The climate scenarios used in this study were made in 2014. Since then, new measurements have become available and models have been improved. KNMI published new scenarios in 2023: four combinations of high/low emissions and a wetting/drying climate (KNMI'23; Van Dorland et al., 2023; Van der Wiel et al., 2024). Unfortunately, the new scenarios were less suitable for this study. First, rainfall projections with hourly resolution are not (yet) publicly available and the extreme hourly

rainfall sums computed with RACMO are quite uncertain (Van Dorland et al., 2023). Second, the time series for KNMI'23 are not based on transformations but on independent runs with the RACMO climate model. The transformations in the KNMI'14 scenarios allowed us to compare flood drivers for the same events. Third, for KNMI'23 eight 30-year time series have been constructed by resampling the RACMO runs: stretches of 1–11 years were cut from one of the sixteen RACMO runs and pasted together. When the eight KNMI'23 ensemble members are then pasted together to obtain a 240-year series (for robust statisti-

cal analyses), there are in total 58 discontinuous December to January transitions. This may be disadvantageous for studies on floods (and droughts), because the memory of the hydrological system partly determines the flood risk. Therefore, we decided to use the older KNMI'14 climate scenarios for this study.

## 2.3 Rainfall-runoff model

The rainfall-runoff model used for this analysis is the Wageningen Lowland Runoff Simulator (WALRUS; Brauer et al.,

2014a). WALRUS accounts for hydrological processes relevant to areas with shallow groundwater, notably (1) groundwater-unsaturated zone coupling, (2) wetness-dependent flowroutes, (3) groundwater-surface water feedbacks and (4) seepage and surface water supply or extraction. We chose WALRUS because its model structure is suitable for the chosen areas, it explicitly simulates groundwater depth (as a catchment-effective value) and runs fast. WALRUS is used by several Dutch water authorities for flood and drought forecasting and offline simulations, with good performance compared to observations (Sun et al., 2020;

Burke et al., 2021; Moekestorm et al., 2025).

WALRUS consists of three reservoirs: (1) a coupled groundwater-vadose zone reservoir, (2) a quickflow reservoir and (3) a surface water reservoir (Fig. 2). It requires rainfall, potential evaporation, and, if applicable, seepage and surface water supply as input.

Rain water is divided between the three reservoirs: a fixed fraction depending on the area of the catchment covered with

surface water ($a_S$) is led to the surface water reservoir and the remainder (the land fraction, $a_G$) is divided between the groundwater-vadose zone reservoir ($P_V$) and the quickflow reservoir ($P_Q$). This division depends on the catchment wetness ($W$), which in turn depends on the storage deficit ($d_V$, the lack of water in the groundwater-vadose zone reservoir) and a model parameter $c_W$. Water can evaporate from the surface water reservoir ($ET_S$) and the groundwater-vadose zone reservoir, where the actual evapotranspiration ($ET_V$) equals the potential evapotranspiration ($ET_{pot}$) multiplied with a reduction factor ($\beta$)

based on the storage deficit.

The groundwater depth ($d_G$) depends on the storage deficit through a soil type dependent relation, and responds to changes in storage deficit with a delay (determined by parameter $c_V$). Water can both flow from the groundwater-vadose zone reservoir to the surface water reservoir and the other way around ($f_{GS}$), depending on the groundwater depth and surface water level



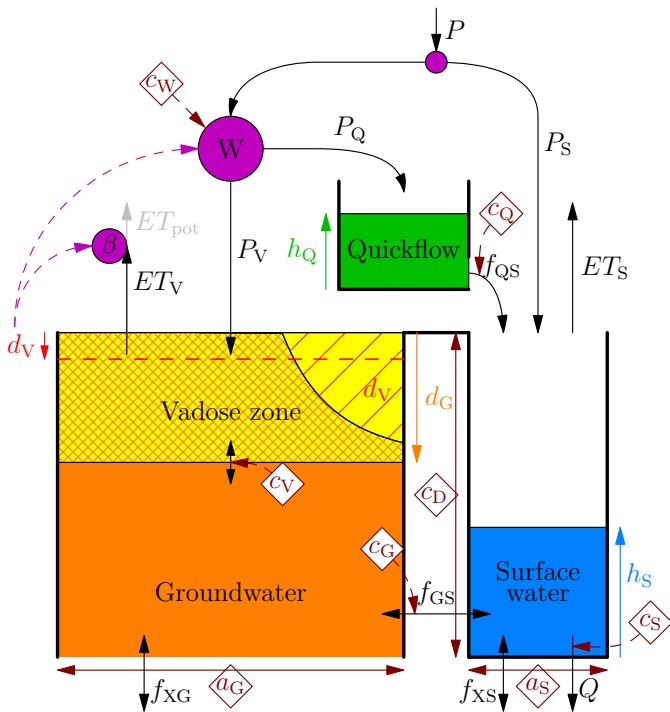

**Figure 2.** WALRUS model structure with the three reservoirs (orange + yellow, green and blue) and their state variables (coloured arrows), fluxes (black arrows) and model parameters (brown diamonds. Figure copied from Brauer et al. (2014a).

($h_S$), the channel depth ($c_D$) and a groundwater reservoir constant ($c_G$). Seepage ($f_{XG}$) is added to or removed from the

groundwater-vadose zone reservoir.

Rain water entering the quickflow reservoir raises the quickflow reservoir level ($h_Q$) and flows to the surface water ($f_{QS}$) using a linear reservoir constant ($c_Q$). Surface water supply ($f_{XS}$) is added to the surface water reservoir. Discharge ($Q$) is the outflow of the surface water reservoir, which depends on the surface water level ($h_S$) and a stage-discharge relation, often using a default relation (with parameter $c_S$).

WALRUS has been applied to all catchments in earlier studies (Tab. 1). We ran WALRUS with hourly resolution because the fastest catchment has a response time of about 3 hours (Brauer et al., 2018). We used different forcings (Sec. 2.2) and different catchment-specific model settings (model parameters and additional forcing, if applicable; Tab. 1). WALRUS parameter values were obtained from previous studies or calibrated using first Latin-Hypercube sampling (McKay et al., 1979) with a sample size of 1000 and varying $c_W$, $c_G$ and $c_Q$, and then using the best 10 parameter sets as input for a Levenberg-Marquardt optimization

(Levenberg, 1944; Marquardt, 1963). We calibrated these three parameters because these are the most sensitive (Brauer et al., 2014b).

The model parameters and the amount and period of seepage and surface water supply have been kept constant for all runs for a specific catchment. It is likely that water management and catchment characteristics will change in the future, which





would lead to different model settings (e.g. Bouaziz et al., 2022), but the direction and magnitude of these changes and their result on WALRUS model parameters and variables is highly uncertain so we decided to limit this study to examining the effect of changes in forcing.

The internal computation time step size of WALRUS is automatically decreased in case of high rainfall or large water level variations within one time step (for more information, see Brauer et al., 2014a).

In this study, we limit ourselves to simulated groundwater depth and discharge, since time series of observations are too short 190 for robust statistical analyses. The simulated groundwater depth is an effective catchment value and represents the seasonal variation rather than the quick responses to rainfall (those are incorporated in the quickflow reservoir). The modelled groundwater depth depends directly on the wetness of the topsoil and can therefore be used as indicator for the catchment wetness. Discharge cannot be related directly to the groundwater level, because water can flow towards the surface water network via fast flowroutes as well. Therefore, the discharge signal is a combination of the slow variation as modelled by the groundwater 195 reservoir and fast variation as modelled by the quickflow reservoir, and as a result it is more variable than the groundwater level.

## 2.4 Discharge dynamics

We computed three discharge metrics to quantify the dynamics for each catchments: slope of the flow duration curve, baseflow index and flashiness index. All metrics are computed on hourly values of the specific discharge (in $\mathrm{mm\,h^{-1}}$). These metrics 200 are computed from simulations rather than observations because the observed time series are too short.

The slope of the flow duration curve is computed between the 33th and 66th percentile of discharge ($Q$) (e.g. Sawicz et al., 2011):

$$\text{Flow duration curve slope} = \frac{\ln(Q_{33}) - \ln(Q_{66})}{0.66 - 0.33} . \tag{1}$$

Steep slopes represent catchments with a large range of discharges.

The baseflow index is the faction of baseflow relative to the total discharge:

$$\text{Baseflow index} = \frac{\Sigma Q_{\text{baseflow}}}{\Sigma Q_{\text{total}}} . \tag{2}$$

The baseflow is estimated using the method by Gustard and Demuth (2009), as implemented in the R package lfstat, which was developed to compute several statistics for low flows (Koffler et al., 2016). A high baseflow index indicates that a large portion of the discharge originates from slowly varying flow routes. In natural catchments this points at a large contribution 210 from groundwater, but in (often human influenced) lowland catchments it can also be caused by surface water supply or upward seepage.

The flashiness index is defined as the ratio of the mean absolute difference between subsequent hourly discharges and the overall mean discharge, computed over the entire series of 109 years (i.e. $n = 109$ years $\times$ 365 days $\times$ 24 hours = 954,840 hourly intervals):

$$\text{Flashiness index} = \frac{\Sigma_{i=1}^{n-1}|Q_{i+1} - Q_i|}{\Sigma_{i=1}^{n-1}Q_i} . \tag{3}$$





A high flashiness index shows that there is much variation between consecutive hours. The flashiness index quantifies short-term variability, whereas the slope of the flow duration curve only accounts for the total variation without considering chronology (Wannasin et al., 2021). For example, a steep flow duration curve slope could be caused by a large seasonal variation or strong responses to rainfall, but only the latter would result in a high flashiness index.

## 2.5 Output analyses

WALRUS was run 108 times (12 catchments × 9 forcing time series) and each run consisted of 109 years of hourly data. Each run took about 30 minutes on an average desktop pc. From this large amount of data, we selected all discharge peaks above a threshold and more than 48 hours apart. These events will hereafter be referred to as floods. The threshold differed per catchment (see right panel in Fig. 3) and was set at a value resulting in, on average, 10 floods per year in the current climate. We also used the thresholds based on the current climate to select and evaluate floods in the future climates. We did not regard the season in the selection of the floods, but for some analyses we split the dataset into subsets per season, using Dec–Feb for winter, Mar–May for spring, Jun–Aug for summer and Sep–Nov for autumn.

The threshold value corresponds to the maximum discharge of the 1090th highest peak in the 109-year time series. We chose this threshold value to keep a large enough number of peaks for statistical analyses while only focusing on the higher (more relevant) ones. For some figures or analyses, we used a subset, focusing on, for example, the 10 % highest peaks (on average one peak per year), or a certain season.

For all floods four metrics were computed (see supplement for an illustration): the effective rainfall sum over the 48 hours preceding the discharge peak ($\Sigma(P-ET)$), the groundwater depth 48 hours before the discharge peak ($d_\mathrm{G}$), the peak discharge ($Q_\mathrm{peak}$) and the volume of discharge above the threshold ($V$). We chose the duration of 48 hours because it is long enough before the peak to capture the rainfall event and catchment response of the slowest catchment and short enough such that it is still related to the peak under consideration. We also performed the analyses for several lag times between 6 and 96 hours, but the results were very similar, so we only present the results for a lag time of 48 hours in this paper. The floods are not always independent, since they may occur during a long wet period or during the recession of a previous flood. We chose to compute the volume above the threshold instead of the volume above the baseflow, because this is a more direct measure of the severity of the flood.

To investigate the sensitivity of peak discharge to effective rainfall sum and initial groundwater depth, we first plotted these against each other and interpolated between the points (with $Q_\mathrm{peak}$ on the $z$ axis) to obtain a surface. We used the method of bivariate interpolation and smooth surface fitting by Akima (1978), as implemented in the R package called akima. Next, we used multilinear regression:

$$Q_\mathrm{peak} = a + b_{P-ET} \times \Sigma(P-ET) + b_{d_\mathrm{G}} \times d_\mathrm{G} \,, \tag{4}$$

where $a$ is the intercept, $b_{P-ET}$ is the slope in the $P-ET$ direction and $b_{d_\mathrm{G}}$ is the slope in the $d_G$ direction. We chose multilinear regression because it is parsimonious and yields relations that are easy to interpret physically.





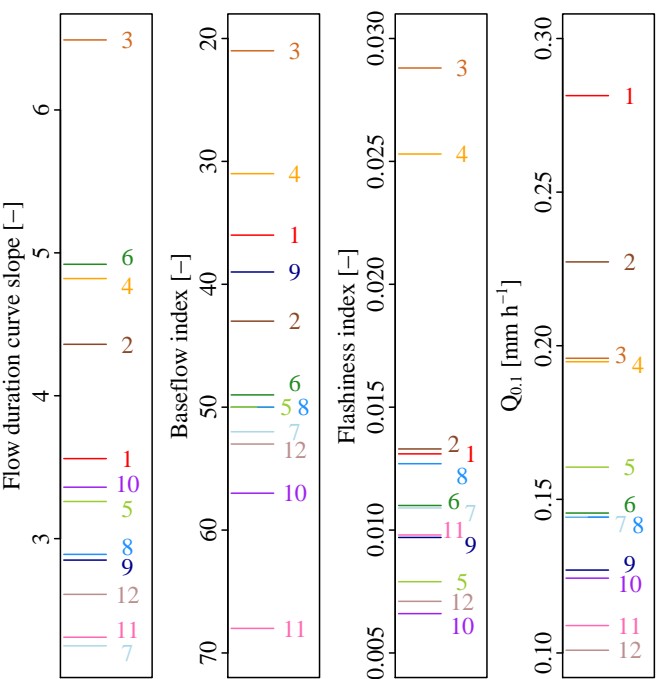

**Figure 3.** Metrics for discharge dynamics and the discharge threshold for all catchments (as numbers; see Tab. 1 for the names and characteristics and Fig. 1 for the locations) leading to on average 10 peaks per year.

## 3 Results

### 3.1 Discharge dynamics

Classifying the catchments as flashy or steady is not straightforward, because it depends on which metric for discharge dynamics is considered (Fig. 3). The four catchments with the highest discharge thresholds (Hupsel Brook (no. 1), Steinfurter Aa (2), Luntersebeek (3) and Aa of Weerijs (4)) are also among the catchments with the steepest flow duration curves, lowest baseflow indices and highest flashiness indices. These catchments are located in areas with more elevation differences and coarser soil material than the other catchments. Surface water supply ($f_{\mathrm{XS}}$, in Ommerkanaal (no. 5), Bakelse Aa (7) and Grote Waterleiding

(11)) and upward seepage ($f_{\mathrm{XG}}$, in Bakelse Aa (7) and Lage Raam (10)) prevent discharges from dropping in summer, leading to high baseflow indices for other catchments.

### 3.2 Conditions leading to floods

In order to understand why and how conditions leading to floods will change with climate change, one should first understand how the various metrics are related for the current climate. In this section, the output from the simulations using the detrended

time series (representative for the current climate) from De Bilt is analysed.





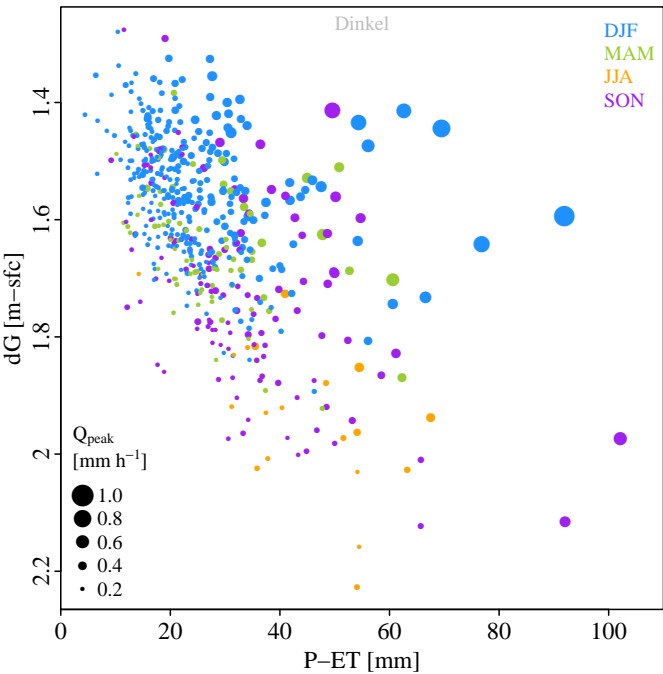

**Figure 4.** Conditions leading to the 50% highest floods (i.e. leading to on average 5 floods per year) in the Dinkel catchment. Each circle represents a flood, with its colour indicating the season, its size proportional to the peak discharge and its location indicating the effective rainfall sum and initial groundwater depth belonging to that flood. Figures for the other catchments are provided in the supplement.

Figure 4 shows which conditions lead to floods with return periods above 0.2 year (i.e. the highest 50% of the 1090 selected floods) for one of the catchments (the Dinkel catchment; plot for the other catchments are similar and given in the supplement). As expected, the sizes of the circles, indicating peak discharges, increase when moving towards the top right corner of the figure, representing high effective rainfall sums and shallow initial groundwater tables.

The colours in Figure 4 indicate the season. Most floods occur in winter (blue circles), when groundwater is shallow and relatively small effective rainfall sums (above 25 mm) can already lead to high peak discharges. In summer, groundwater is deep and floods only occur when the effective rainfall sum is high. The same holds to a lesser extent for autumn.

### 3.2.1    Sensitivity of peak discharge to effective rainfall sum and initial groundwater depth

Figure 5 gives the surfaces obtained by interpolating between the points from Figure 4 for two (other) contrasting catchments.
Steinfurter Aa (no. 2) is one of the catchments with relatively hilly terrain and shallow aquifers, resulting in relatively flashy hydrographs (Fig. 3). Ommerkanaal (no. 5) is one of the catchments with hardly any topography and a larger buffering capacity of the soil, resulting in more gradually changing discharges and low flashiness index.

These differences also impact the occurrence and magnitude of floods. Peak discharges are higher in the Steinfurter Aa (up to $1.2 \, \mathrm{mm \, h^{-1}}$) than in the Ommerkanaal (up to $0.84 \, \mathrm{mm \, h^{-1}}$). These high discharges in the Steinfurter Aa occur after



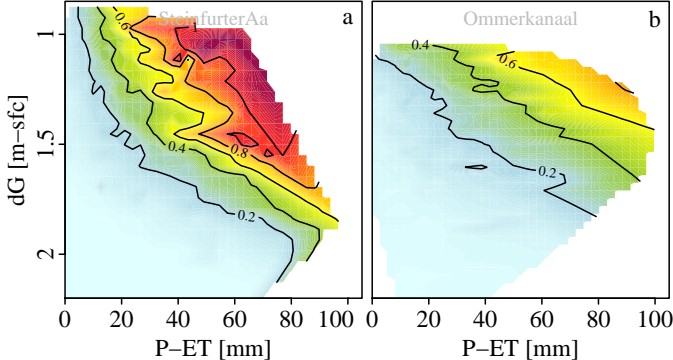

**Figure 5.** Sensitivity of peak discharge (colour and contour lines in mm h$^{-1}$) to effective rainfall sum ($x$-axis) and initial groundwater depth ($y$-axis) for two contrasting catchments. The surface is interpolated through the cloud of all floods. Figures for the other catchments are provided in the supplement.

moderately high effective rainfall sums. For the Ommerkanaal, initial groundwater depth has a larger effect on the discharge peak than for the Steinfurter Aa, which is shown by a steeper slope in the vertical ($d_{\mathrm{G}}$) direction.

### 3.2.2 Multilinear regression

In Figure 5 the different slopes can already be identified by eye. To compare these slopes for all combinations of catchments and forcing data sets, we fitted a plane through the points of Figure 4 using the multilinear regression explained in Section 2.5. The resulting parameters $b_{P-ET}$ (sensitivity to effective rainfall sum) and $b_{dG}$ (sensitivity to initial groundwater depth) for all catchments and all scenarios are shown in Figure 6. As an example, the Steinfurter Aa catchment has a steeper slope in the $P - ET$-direction in Figure 4, resulting in a higher value of $b_{P-ET}$ in Figure 6 (brown points) compared to the Ommerkanaal (light green points).

The planes are quite well able to describe the points, with $R^2$ values ranging from 0.69 to 0.84. Parameter $b_{dG}$ is negative, because groundwater is expressed as a depth below surface, resulting in high peak discharges when $d_{\mathrm{G}}$ is low. There is a negative correlation between the two slopes, indicating that a steep slope in the $P - ET$ direction coincides with a steep slope in the $d_{\mathrm{G}}$ direction. Hence, catchments sensitive to the effective rainfall sum are also sensitive to initial groundwater depth. These are the catchments with much variation in discharge and sharp discharge peaks.

The four catchments with the most flashy hydrographs (Hupsel Brook, Steinfurter Aa, Lunterse Beek and Aa of Weerijs) are more sensitive to effective rainfall sums than the other eight catchments. The Hupsel Brook catchment stands out, probably because of its small size (6.5 km$^2$) and shallow aquifer ( 0.2–10 m; Brauer et al., 2018).




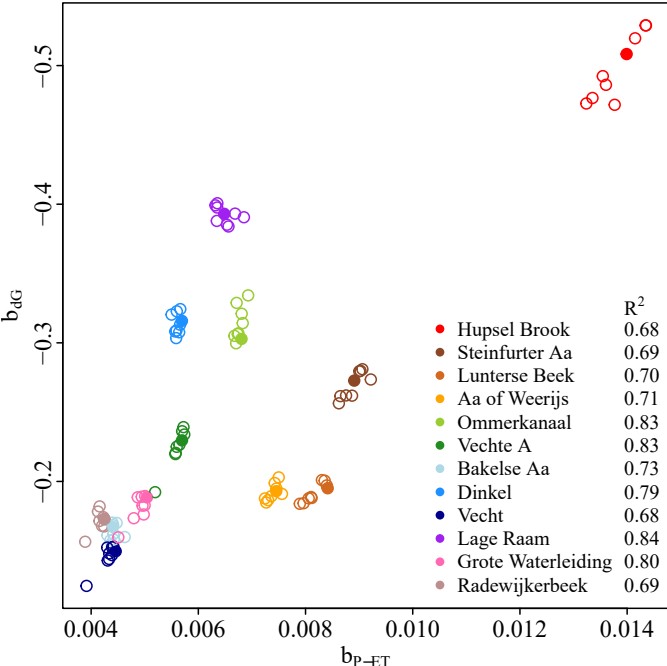

**Figure 6.** Parameters $b_{P-ET}$ and $b_{dG}$ from Eq. 4, together with the corresponding coefficients of determination ($R^2$) for all catchments and all scenarios. The filled circles are for the current climate and the open circles for future climates.

## 3.3 Flood occurrence with climate change

In Fig. 6 there is little difference between the regression parameters fitted on the output using the climate scenarios (open circles). This is not surprising since the relation between effective rainfall sum, initial groundwater depth and peak discharge is determined by catchment characteristics, represented by the model parameters, which were kept constant.

### 3.3.1 Average changes

Generally, the four scenarios agree that the average monthly effective rainfall sum is projected to be higher from November to May and lower from July to September (dark blue and red areas in Fig. 7a). For June and October, scenarios disagree (both blue and red). The WH scenario is the most extreme and projects the largest increase in winter precipitation and summer evapotranspiration, leading to larger differences in effective rainfall sum. Two scenarios show a deviating pattern: the GL scenario projects less winter precipitation and the WL scenario projects more rainfall in May and June than in the current climate.

The average groundwater depth responds to changes in effective rainfall with a delay. More effective rainfall in autumn, winter and spring and less in summer results in shallower groundwater in winter and spring and deeper groundwater in summer and autumn (see Fig. 7b for one catchment). The effect of climate change on groundwater depth is similar for all catchments,





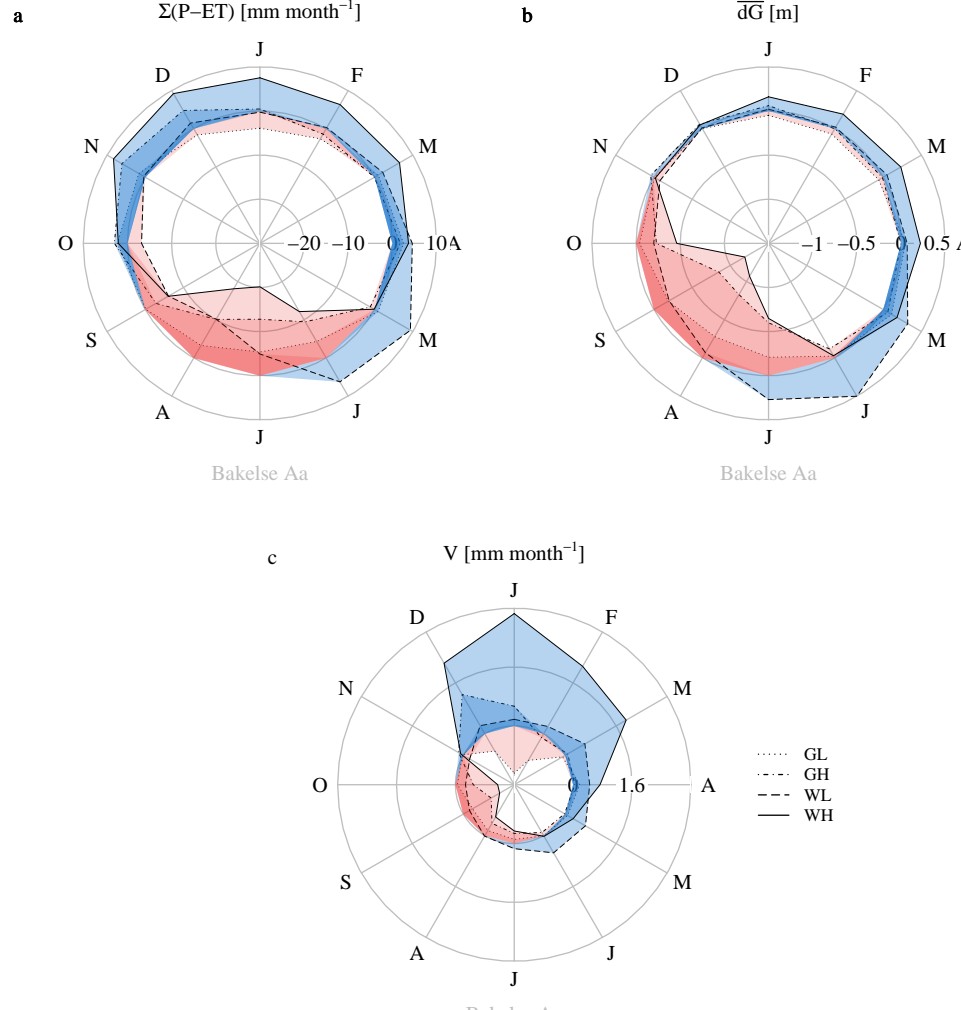

**Figure 7.** Change in **a** effective rainfall sum, **b** groundwater depth and **c** peak volume for 2050 for the Bakelse Aa catchment. Blue means wetter and red means drier with respect to the current climate. The different scenarios are overlaid, so a darker hue means that scenarios agree. Figures for the other catchments are provided in the supplement.

though the speed with which catchments recover from deeper groundwater in summer varies. For most catchments, groundwater in November is expected to be still a bit deeper than in the current climate, but in the Vecht, Aa of Weerijs, Luntersebeek, Lage Raam and Radewijkerbeek (all catchments with a low value of $b_{\mathrm{d_G}}$ in Fig. 6), groundwater in November is expected to be still considerably deeper. For one scenario in the Dinkel, the wet season starts earlier and groundwater is already shallower

in November compared to the current climate (see supplement).

The combined effect of effective rainfall and groundwater depth leads to a larger discharge peak volume from December to June and a smaller peak volume from July to October (Fig. 7c). The GL scenario is the only exception: lower effective rainfall





sums cause smaller peak volumes in winter. The largest increase in peak volume in winter and decrease in peak volume in summer is projected for the WH scenario.

### 3.3.2   Changes for individual floods

The fact that the time series corresponding to the four climate change scenarios are transformations of the original time series allows us to investigate how the conditions leading to individual floods would change as a result of climate change if the catchment characteristics would remain constant (Fig. 8). Under the assumptions of this modelling study, 98 % of the 1090 floods in the current climate would also occur in future scenarios within 24 hours before and 24 hours after the discharge peak in the current climate (averaged over all catchments and all climate scenarios), using the thresholds determined on the current climate. For the the remaining 2 %, the transformed effective rainfall sum was too small or the initial groundwater too deep to cause a discharge peak that exceeded the threshold.

Blue arrows in Fig. 8 mostly point to the right, indicating an increase in effective rainfall sum and little change in initial groundwater depth in winter. Green arrows mostly point to the top-right, indicating both higher effective rainfall sums and shallower groundwater in spring. Orange arrows are few in number and point down, indicating little change in effective rainfall sum and deeper groundwater for the few floods that reach the threshold in summer. Purple arrows point to the bottom-right, indicating higher effective rainfall sums but deeper groundwater in autumn after drier summers. Since peak discharges increase when moving to the right and top of the graph (as seen in Fig. 4), spring and winter floods would intensify, but summer and autumn floods would decrease in magnitude.

Averaging the changes over all floods per season and scenario leads to Figure 8b (for one catchment). The values are small: only several millimeters more or less effective rainfall sum and up to 14 cm initial groundwater depth change, but the combined effect on peak discharge and flood volume can be significant. Changes are smallest for the GL scenario, which represents moderate changes in temperature and atmospheric circulation patterns. The scenarios with larger changes in circulation patterns, GH and WH, result in the largest changes in initial groundwater depth. The WH scenario leads to the largest changes: higher effective rainfall sums in autumn, winter and spring, deeper groundwater in summer and autumn and shallower groundwater in spring. Note that the WH scenario for 2085 projects higher effective rainfall sums in summer than the WH scenario for 2050, which results in less decrease in peak discharges. The WL 2050, WL 2085 and GL 2085 scenarios project an increase in effective rainfall sum in summer and therefore higher peak discharges.

Averaged over all scenarios and all catchments, the effective rainfall sum increases with 0.5 mm per event (1.5 %) and the initial groundwater depth increases with 11 mm (0.7 %) for 2050. For 2085, the effective rainfall sum increases with 1.5 mm per event (5.6 %) and the initial groundwater depth deepens with 7 mm (0.3 %). The deeper initial groundwater tables partly counter the effect of higher rainfall sums, reducing the increase in number and severity of severe floods in the future.

### 3.3.3   Differences between scenarios and catchments

Averaged over all catchments, scenarios and seasons, floods become more frequent and severe (Fig. 9). To analyse the difference between minor and more severe floods, we used thresholds which led to on average 10, 1 or 0.1 floods per year in the





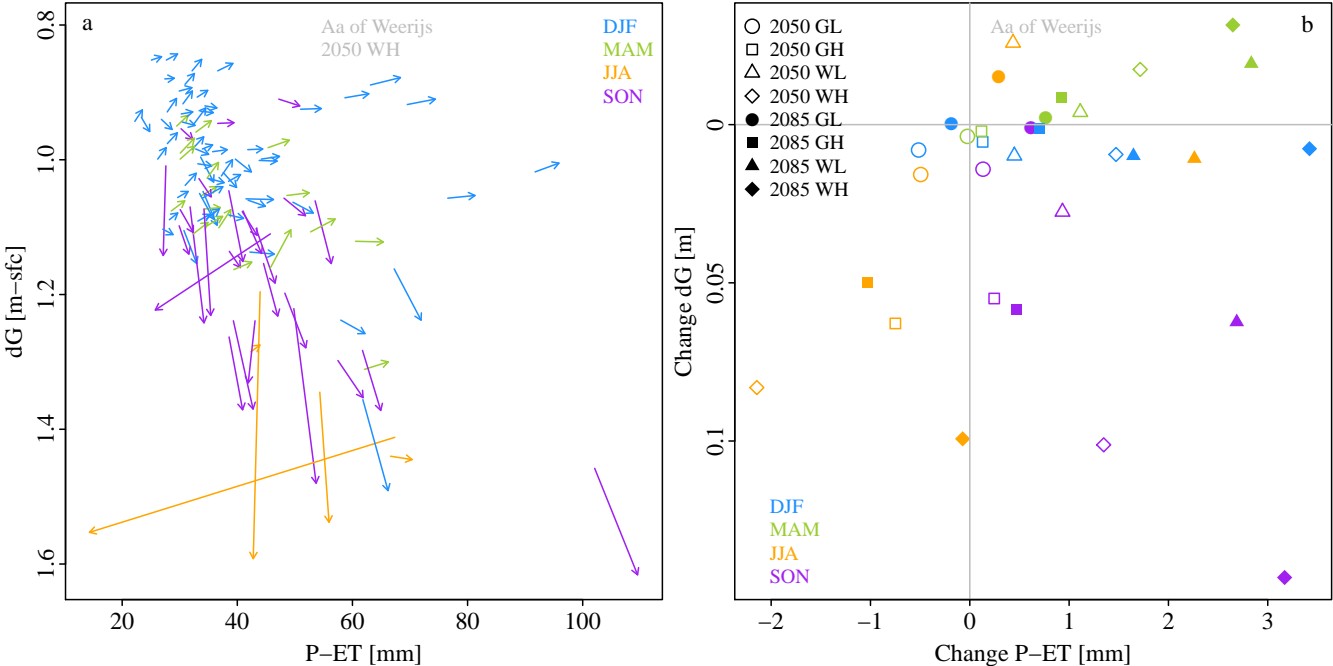

**Figure 8.** Change of conditions leading to floods in the Aa of Weerijs catchment. **a** Effective rainfall sum and initial groundwater depth for each flood in the current climate (start of arrows) and in the WH 2050 scenario (end of arrows). Only floods with return period of 1 year and higher are shown here. **b** The arrows of the left figure (and those for lower return periods, but above the predefined threshold) averaged per season and per scenario. For example, all orange arrows in the left panel are averaged to the open orange diamond in the right panel. Figures for the other catchments are provided in the supplement.

current climate (return periods of 0.1, 1 and 10 years, respectively). For minor floods (i.e. using a threshold of 10 peaks per year, as we did in the previous sections), the change is small in 2050 (1 % more floods and 3 % larger total peak volume), but more substantial in 2050 (9 % more floods and 21 % larger peak volume).

Differences between scenarios are larger than between catchments. For minor floods, averaged over all catchments, the GL
and GH scenarios only lead to 3% fewer floods and 4% less peak volume in 2050 and 3 % more floods and 6 % more volume in 2085. The WL and WH scenarios, however, lead to 5 % more floods and 10 % more volume in 2050 and 15 % more floods and 36 % more volume in 2085. Differences between catchments are most visible in the changes for 2085 WH, for which also the largest absolute changes are projected. The Radewijkerbeek exhibits the largest increase in number of floods (up to 222 % for severe floods in 2085 WH).

For most catchments, the changes are larger when considering more severe floods: the extremes become more extreme. Averaging over all catchments and scenarios, the number of floods is expected to increase with 1 %, 6 % and 18 % in 2050, and 9 %, 31 % and 57 % in 2085, where the three values correspond to thresholds of 10, 1 and 0.1 floods per year, respectively. The total peak volume show a similar pattern, increasing with 4 %, 10 % and 20 % for 2050, and 20 %, 45 % and 60 % for 2085.





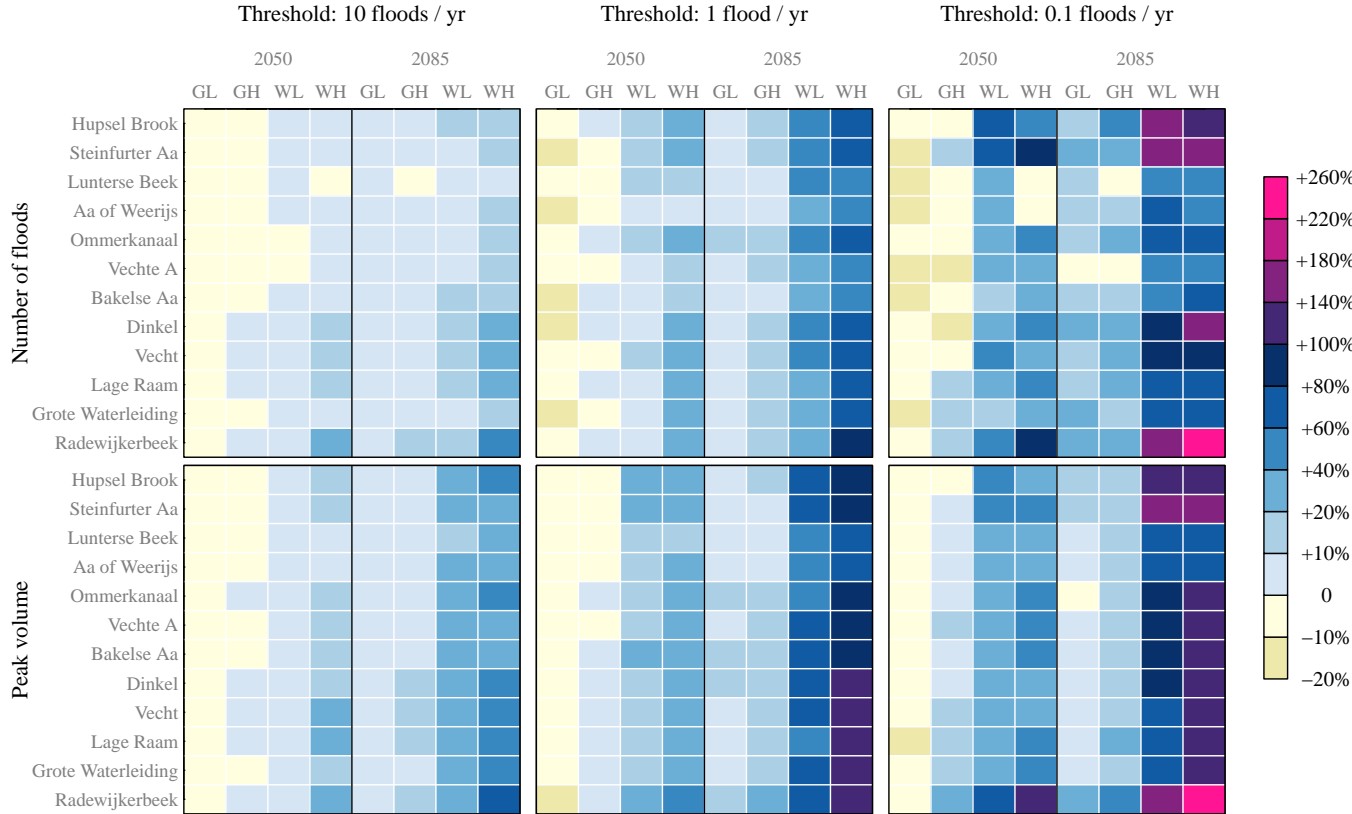

**Figure 9.** Change in the total number of floods (top row) and peak volume (bottom row) per catchment and per scenario, compared to the current climate. We distinguish between minor and more severe floods: floods are defined as exceedances of the threshold which led to on average 10 (left column), 1 (middle column) and 0.1 (right column) floods per year in the current climate.

# 4 Discussion

## 4.1 Limitations of the study

To project future floods, we used different forcing data, but kept the rainfall-runoff model, its parameters and additional input variables (seepage and surface water supply) the same. Climate adaptation through changes in land use and water management are therefore not accounted for. Dutch water managers are aware of the increasing flood risk, and implementing measures at different scales and with different techniques to reduce the impact of climate change has a high priority (e.g. Bartholomeus 365 et al., 2023). This adaptation to changing conditions could result in lower peak discharges and volumes than projected in this study. In addition to human adaptation, natural vegetation could change to adapt to changing climatic conditions (e.g. Bouaziz et al., 2022).




In the WALRUS model, the partitioning of rain between quick and slow flowroutes depends only on the catchment wetness. Infiltration excess overland flow is not incorporated in WALRUS because this is assumed to be of limited importance on regional scale in areas with limited topography. Infiltration excess occurs locally, but often stays on the fields in local depressions and infiltrates later (Appels, 2013). Though Schaap et al. (2024) points out the relevance of overland flow for water quality management, the importance for flood peaks at the catchment outlet remains unclear because measuring overland flow in lowland fields is challenging. In addition, simulating infiltration excess overland flow is not trivial since it is affected by the temporal resolution of the rainfall data, which does not match with WALRUS' flexible time step approach. However, it is possible that the projected increase in rainfall intensity in summer will lead to more infiltration excess overland flow and therefore to higher peak discharges and peak volumes.

As in every simulation study, there is uncertainty caused by the model parameters. In a previous study we examined the effect of parameter uncertainty and other uncertainties in the WALRUS model in detail (Brauer et al., 2014b). For the current study, the simulations were validated, both quantitatively by comparing observed and simulated discharges and qualitatively by assessing the realism of internal model variables. The difference in results between the catchments already gives an indication of the spread caused by different parameter values. The comparable results between the catchments suggests that it is unlikely that parameter uncertainty would affect the general conclusions of this study related to the importance of initial wetness for flood generation and expected changes, though it may affect the exact percentages of expected changes in flood volumes and number of floods (presented in Fig. 9).

Using the same forcing for each catchment is a simplification of reality, since there are small differences in climate over the Netherlands and neighbouring regions of Germany and Belgium. However, identical forcing allows us to analyse the relation between initial wetness and floods and the effect of climate change as function of catchment characteristics (or response behaviour) in isolation. This is especially difficult to distinguish in studies using observed discharges. Lowland areas are often located in river deltas, with both maritime climates and limited topography. In studies comparing flood changes between a large number of catchments (e.g. Hall et al., 2014; Blöschl et al., 2017; Berghuijs et al., 2019), it is not known if the similarities in observed changes in a certain region are caused by similarities in climate or landscape.

The KNMI'14 climate scenarios are not simply multiplications of the original observations by a fixed factor – dry spells have been expanded or shortened, rainfall events have been split or combined, and intensities were increased or reduced with variable factors, such that the rainfall statistics mimic the climate model output as closely as possible in terms or intensity, duration and volume (Beersma et al., 2015). However, this transformation method may still miss changes caused by changing circulation patterns. To assess the effect of the chosen methods, Manola et al. (2018) compared three projections for a summer event: a delta change method (based on the values of the KNMI'14 scenarios), dewpoint-scaling and the HARMONIE numerical weather prediction model for single summer event. They found that HARMONIE predicted rainfall earlier in the day while the other two methods did not shift the timing of the event, but rainfall amounts and coverage were similar. Hence, we expect that the conclusions of this study will not change much for the studied lowland catchments when a different climate forcing method would be applied.





## 4.2 Comparison with previous studies

The conclusions resulting from this study can be partly extrapolated to other lowland catchments worldwide. The effect of storage depletion caused by evapotranspiration on the response to rainfall later in the season will be similar in many areas, but the climate projections and therefore the resulting shifts in hydrological processes are site-specific. Special care should be taken when exporting the technique to snow impacted catchments. Snow is of limited importance in the Netherlands (only 1 % of the annual precipitation falls as snow in the current climate; Brauer, 2014; Brauer et al., 2018), so we did not use the module for snow accumulation and melt implemented in the WALRUS model. For the catchments used in this study, peak discharge can be well explained with only effective rainfall sum and initial groundwater depth (see $R^2$ values in Fig. 6). However, in many lowland areas worldwide, snowmelt is an important driver of floods (Liu et al., 2022) and would need to be added as a third explanatory variable.

The result that events with both shallow groundwater tables and large rainfall sums cause higher floods is not surprising. Jiang et al. (2024) identified flood drivers for many events and found that 52% of the identified flood events are partly explained by soil moisture and that more extreme floods occurred in situations where multiple drivers (rain, snow, temperature and soil moisture) played a role. Before this study we did not know how large the compound (or trade-off) effect of groundwater and rainfall would be in the Netherlands. We found that groundwater plays a large role in determining the flood severity, both as a factor mitigating floods in fall and aggravating floods in spring.

The prevalence of floods during periods of high soil wetness (in particular in winter) that we found corresponds to earlier studies in western Europe and Germany (Tarasova et al., 2020b, a). They found an increase in number of floods after wet initial conditions (Tarasova et al., 2023; Tsiokanos et al., 2024). In our study, we found that initial groundwater tables became either a little shallower (spring) or much deeper (summer and fall; see Fig. 8). However, due to the nonlinear processes leading to discharge response, the resulting floods after somewhat shallower groundwater in spring led to significantly higher flood volumes (Fig. 7).

The shift of winter floods towards spring found in this study corresponds to the time shift observed in historical data in the North Sea region (Blöschl et al., 2017). Our study points out that this is more caused by the seasonal dynamics of soil wetness than by the increase in precipitation directly, since precipitation is also projected to increase in fall (see Fig. 7).

Our finding that severe floods increase more than minor floods is in line with the conclusion of Gründemann et al. (2022) that extreme precipitation is projected to become more extreme. Bertola et al. (2020) found the same for small ($<100$ km$^2$) catchments, but the opposite for larger catchments (up to 100,000 km$^2$). The largest catchment we considered (Vecht) has a surface area of 2821 km$^2$ and showed the same pattern as the smaller ones. The fact that we used the same forcing for all catchments, which was based on point measurements and not adjusted for the areal reduction effect, may have played a role here.

Compared to the KNMI'14 scenarios, the recently released KNMI'23 scenarios project similar changes for the wet scenarios, less rain for the dry scenarios (less increase in spring and autumn and stronger decrease in summer) and less rain in summer for all four scenarios (instead of two scenarios in KNMI'14). We expect that this would result in fewer and lower



floods in summer and autumn, caused by the higher precipitation deficit, and thereby lower initial wetness in summer. Buitink et al. (2023) found similar results for for the Rhine and Meuse for the KNMI'23 scenarios: higher discharges in winter and spring and lower in summer and early autumn.

## 5 Conclusions

The aim of this study was to determine the importance of initial wetness on flood peaks in lowland catchments and to examine if and how this affects the magnitude and timing of floods in the future. We used 109 years of hourly precipitation and evapotranspiration data for the current climate and eight climate scenarios to simulate discharge and groundwater depth with the rainfall-runoff model WALRUS, which is designed for lowlands, performs well and is used in practice. Then we investigated the relation between initial groundwater depth, effective rainfall sum and the resulting peak discharge and peak volume for 12 lowland catchments. We found that this relation is strong in these catchments and that the highest flood peaks can often be attributed to the simultaneous occurrence of shallow groundwater levels and high precipitation amounts, which would allow water managers to better estimate peak discharges based on the initial groundwater depth and weather forecasts. We parameterized the effects of rainfall and groundwater on peak discharge and found that catchments without topography were more sensitive to groundwater depth than catchments with some elevation differences.

When climate changes, less precipitation and more evapotranspiration is projected in summer, resulting in deeper groundwater in summer and autumn, reducing flood occurrence and magnitude. More rain in autumn, winter and spring will lead to more frequent and more severe floods in winter and spring only, because in autumn groundwater is still recovering from summer, counteracting the effect of more rainfall. Averaged over all scenarios and all catchments, the effective rainfall sum increases with 1.5 % in 2050 and 5.6 % in 2085, while the initial groundwater depth deepens with 0.7 % in 2050 and 0.3 % in 2085. Without the mitigating effect of the deeper initial groundwater tables, the higher rainfall sums would have led to more frequent and more severe floods in these lowland catchments in the future.

Differences between climate scenarios are found to be larger than differences between catchments. Averaged over all catchments, scenarios and seasons, floods that currently occur 10 times per year, are projected to become more frequent and severe, from 1 % more floods and 3 % larger total peak volume in 2050 to 9 % more floods and 21 % larger peak volume in 2085. This increase is projected to be stronger for more extreme events.

The knowledge that the initial groundwater depth affects the expected changes in flood seasonality and severity can be used to design climate robust water management in lowlands in delta areas worldwide. On the one hand, the strong dependence on groundwater depth makes lowlands more vulnerable to floods, but on the other hand this sensitivity to groundwater depth offers opportunities to reduce flood risk by storing and discharging water at the right moment or by making efficient use of inflow from upstream areas and reservoirs in dry periods. More flexible surface water and groundwater level management, which requires changes in land use and surface water network combined with accurate forecasts, could help to store water in wet periods and release it in dry periods and thereby alleviate floods and combat droughts.



*Code and data availability.* The WALRUS model can be downloaded from www.github.com/ClaudiaBrauer/WALRUS. The KNMI'14 time
series can be downloaded from www.meteobase.nl.

*Author contributions.* CB: conceptualization, formal analysis, methodology, visualization, writing - original draft preparation, writing - review & editing. RI: conceptualization, methodology, writing - original draft preparation, writing - review & editing. RU: conceptualization, methodology, writing - review & editing.

*Competing interests.* No competing interests are present.

*Acknowledgements.* We thank Danny Heuvelink for his contribution to the first analyses. We thank Jules Beersma (KNMI) for providing the transformed time series of precipitation and potential evapotranspiration and all water authorities for the discharge data, and in some cases WALRUS parameter estimates.



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
