# Peer review of "Rain-on-wet-soil compound floods in lowlands: the combined effect of large rain events and shallow groundwater on discharge peaks in a changing climate"

_EGUsphere, 2025_

## Author Comment (AC1)

*Response to reviewer's comments after the submission of "Rain-on-wet-soil compound floods in lowlands: the combined effect of large rain events and shallow groundwater on discharge peaks in a changing climate" by Claudia Brauer, Ruben Imhoff and Remko Uijlenhoet, submitted to Hydrology and Earth System Sciences.*

We thank both reviewers you for their constructive remarks. In the text below we respond to each item.

**Reviewer 1**

In the manuscript "Rain-on-wet-soil compound floods in lowlands: the combined effect of large rain events and shallow groundwater on discharge peaks in a changing climate" authors modeled flood response in 12 lowland catchments in the Netherlands and studied the relationship between effective rainfall, groundwater depth, and flood volume. In addition to studying current conditions, an observed climate forcing dataset was transformed to mimic future climate scenarios, and the change in the rainfall, depth, and flood relationship was evaluated as well.

The study finds that high effective rainfall in combination with shallow groundwater table depth led to higher flood flows. This relationship is consistent across all catchments, although flashy catchments are more sensitive to effective rainfall. The dependence on groundwater depth means that seasonal changes in groundwater depth, such as drying during the summer months leads to a delayed reduction of flood peaks in fall, despite increased effective rainfall. In winter and spring groundwater depth recovered and lead to higher peak volume. These relationships are similar under future climates. The study mostly analyses frequent flood events (10 per year), but also studies rare events (0.1 per year). These rare, most extreme events are increasing in frequency the most under future scenarios with the predicted increases similar between catchments.

Overall, I find the focus on lowland catchment and the role of groundwater depth in flooding very relevant, as it has not been studied in detail much so far in Europe. Since groundwater depth can be measured much easier knowledge about the relationship to flooding can advance flood prediction. However, there are a few open questions that need to still be addressed.

**General comments**

All sections: The authors oscillate between the terms "(initial) catchment wetness", "groundwater depth", and "soil wetness" (e.g. Section 4.2). These terms are not interchangeable and should not be used as such. Especially soil wetness is not equal to groundwater depth and should not be used as synonym. While you define at the end of section 2.3 that you assume groundwater depth to be representative of topsoil wetness, conceptually these are different terms and need to be more clearly defined early on if you are using them in a different context. I would argue to continuously use groundwater depth. Although groundwater depth depends on soil wetness within the model, the variables still mean different things.

Thank you for pointing this out. We agree that it is important to avoid confusion. We will adapt the text to make it consistent. Regarding our own study, we intended to (and will) use groundwater depth throughout the paper, while indeed emphasizing that there is a strong link between groundwater depth and topsoil wetness.

Abstract: The first sentence states that the severity of floods is determined by initial wetness, while the next sentence (and the entire study) then declares that this relationship needs to be studied more. It should be more explicit, that the relationship between soil moisture and floods is well studied, but between groundwater depth and floods has not, especially in lowland catchments.

We agree that your suggestion is better and will change the first sentences of the abstract to: "The relationship between initial soil moisture and floods is well studied in sloping areas, but not in lowland catchments, where the saturated zone, unsaturated zone and surface water are strongly coupled. The aim of this study was to determine the importance of initial groundwater depth (representing soil wetness) on flood peaks in lowland catchments and to examine if and how this affects the magnitude and timing of floods in the future."

Section 1: The summary of previous flood trend and flood type studies is very long. Please consider if you can make the three paragraphs more concise or summarise the findings in a table. I would also recommend focusing more strongly on previous studies that show a relationship between groundwater and flooding, e.g.

Berghuijs, W. R., & Slater, L. J. (2023). Groundwater shapes North American river floods. Environmental Research Letters, 18(3), 034043.

Shamsudduha, M., Taylor, R. G., Haq, M. I., Nowreen, S., Zahid, A., & Ahmed, K. M. U. (2022). The Bengal water machine: quantified freshwater capture in Bangladesh. Science, 377(6612), 1315-1319.

We will shorten the literature review and make it more focused. Thank you for the reference suggestion.

Section 2.1: The results of the study rely on a well-performing model. Currently, the reader is given no information besides the declaration by the authors that the model was validated. Besides the location of the catchments, no information is given in regard to the observed discharge time series that must have been used both to calibrate the model for some catchments and that have been used for evaluation. How long are the time series? Are they at an hourly resolution and was the model evaluated at an hourly resolution? How well did the model perform compared to observed values? What evaluation measure is used? Although the authors mention in Section 4.1 (Line 379) that the model was validated for this current study, no validation information has been given. Against what was the model validated? Only discharge or groundwater depth measurements as well, as the model performance in that regard is quite relevant for this study? What does it mean when you say model simulations were validated "qualitatively by assessing the realism of internal model variables" (line 380)?

The calibration period and process varied per catchment. The footnote in Table 1 contains references to the documents in which the calibration procedure was explained for the eight catchments we did not calibrate ourselves for this study. We will expand the explanation of the calibration of the four catchments which we re-calibrated for this study in the main text. In addition, we will add an extensive description of the calibration and validation of each catchment in the supplementary material, including time series of the calibration and validation.

For all calibrations and validations, the temporal resolution was hourly and the periods were at least one full year (we will include the exact length in the revised manuscript). For the calibration objective we, and the other people who calibrated the models, used a standard performance metric (Nash-Sutcliffe Efficiency of the discharge) in combination with expert judgment of the internal model variables, to exclude model parameter sets which yielded high Nash-Sutcliffe Efficiencies with unrealistic process representation (such as leading all water through the quickflow reservoir with a low reservoir coefficient, or having surface water infiltration conditions year-round).

Comparing the groundwater depth simulated in WALRUS to field observations is not trivial, since the groundwater reservoir in WALRUS mimics the seasonal variation in groundwater table rather than the fast groundwater response to rainfall events. In addition, the spatial representation is different: catchment average/effective value for the model versus point value for the observations. In a previous study (Brauer et al., HESS, 2014), we compared the groundwater simulations in the WALRUS model to groundwater level observations for two catchments, including the Hupsel Brook catchment, which was one of the twelve catchments in this study. Groundwater levels measured in piezometers which were located in an area of the catchment with a relatively thick unsaturated zone (i.e. more than 2 m) corresponded well to the simulated groundwater table. Groundwater levels measured in piezometers which were located close to the surface water network had very shallow groundwater tables and showed more variation than simulated groundwater tables. In WALRUS, this fast temporal variability is included in the quickflow reservoir, since this fast filling and draining points to fast drainage mechanisms, in particular drainpipe flow and macropore flow. We agree that it is important to explain this in more detail and will include more justification, including a reference to our earlier study and a discussion related to the comparison of groundwater observations and simulations in the paper.

It is unclear why some catchments have their model parameters calibrated while others use previously determined parameters. Were there no prior parameters for the calibrated catchments available? Does it make a difference in performance if a model has been calibrated or used prior determined parameters?
The model was calibrated for all catchments, but for some catchments that was already done in either previous studies or by others for operational purposes (see references in Table 1). We did check the performance of the model for all twelve catchments. The four catchments we calibrated ourselves were either not calibrated yet, or the calibration performed by others did not meet our quality standards (i.e. the parameter values led to unrealistic process representation).

Table 1 is apparently sorted by discharge threshold. It would be helpful to have that value given as a column as well.
The threshold values are shown in the right panel of Figure 3. We will expand the caption of Table 1 to guide the reader there.

Section 2.2: I do not find the abbreviations for the climate scenarios very intuitive. This might be my preference, but since they are only explained once and then continuously used throughout the rest of the paper, can they be slightly expanded (e.g. mod/low, mod/high, warm/low, warm/high or something similar?)
The abbreviations are the formal abbreviations given to the scenarios by the Royal Netherlands Meteorological Institute. To avoid confusion, we kept those names. We do understand your confusion, so we will add explanations to the captions of Table 2 and Figures 7, 8 and 9.

Section 2.4: Why do you compute baseflow using the approach by Gustard and Demuth? Is the modelled flow between the groundwater-vadose zone reservoir to the surface reservoir (Gs) not a representation of baseflow?
We chose the baseflow index (BFI) computation methods by Gustard and Demuth to connect our study to existing literature. The flux from the soil reservoir to the surface water reservoir (fGS) computed by WALRUS is indeed a measure of baseflow and should represent a similar process as the BFI. We computed the ratio between the sum of the flow from the groundwater reservoir to the surface water reservoir and the sum of the discharge, and compared them to the BFI (see figure below). For most catchments, the relation between the

two metrics is strong. Ommerkanaal is probably an outlier because this catchment receives a large amount of surface water (0.5 mm/d during 6 months, so about 183 mm/y). This results in stable discharges during summer which are not caused by groundwater drainage but by a steady inflow of surface water.

We agree that using the contribution of fGS to the total discharge is a better metric to quantify the importance of groundwater drainage and decided to replace all mentions of the BFI with the fGS/Q-ratio for the revised manuscript. Since the relation between BFI and fGS/Q-ratio is quite strong, this does not alter the conclusions.

[Figure]

Section 3.3: How do the findings on flood occurrence with climate change relate to the catchment properties, such as the discharge dynamics (Figure 3), or the special conditions mentioned for five catchments (upward drainage, supplied surface water, line 100-101)?
We will include two figures in the supplementary material to investigate this. In the first figure, we will plot the parameters of the fitted surface presented in Figure 6 against the discharge dynamics metrics in Figure 3. In the second figure, we will plot the percentage increase or decrease in number of floods and flood volume presented in Figure 9 against the discharge dynamics metrics in Figure 3 (with the fGS/Q-ratio instead of the BFI).

Why does the Radewijkerbeek catchment show a much larger increase in frequency and peak volume of floods in the WH scenario?
The increase in flood frequency and flood volume is larger in the WH (and to a lesser extent WL) scenarios for all catchments. The Radewijkerbeek shows the largest increase, but it is not much larger than some other catchments. We checked if there are peculiarities in the simulated discharge time series of the Radewijkerbeek, but this was not the case – there were simply more high peaks in the WH (and WL) scenarios. For the highest threshold (based on 0.1 peak on average per year in the current climate) and the 2085WH scenario, we also found a large increase for the Steinfurter Aa and Dinkel. Because the values for the high threshold are only based on a few events (10 events in the current climate), a few events more or less results in a large percentage increase/decrease.

**Technical comments**

L18-19: check sentence structure. Some duplication.
We fixed it.

Figure 3: No colour is necessary in this figure as every line is labelled. There is no explanation for the meaning of the colour given anyway.
We agree that the colors in Figure 3 are not strictly necessary, but it does make it easier to match the lines and the corresponding labels. It also makes it easier to match the different subcatchments between the four subfigures and with the points in figure 6. In addition, in our opinion, the colors make the figure more attractive. In the revised manuscript, we will explain the meaning of the colors and refer to Figure 6.

All figures with seasonal depictions: The chosen colours for the seasons cannot be read by people with colour vision deficiency. Please think about using alternative colour scales (Stoelzle & Stein 2021)
We will change the colors such that they are distinguishable by people with colour vision deficiency.

Figures 4, 5, 7, 8: You chose to focus on a specific catchment in each of these figures, but it is always a different one. One improvement would be to have one constant example catchment in combination with a chosen catchment that illustrates the point you are trying to make.
We indeed picked for each figure a catchment which was representative or showed a clear behaviour. We put the figures for all catchments in the supplement and referred to it in the captions of the figures in the main text. We thought about including the same catchment in all figures as well, but this would make the paper much longer and less organized.

Figure 7: While a circular diagram is technically correct for a depiction of year round values, the zero point is very difficult to see. Furthermore, it is unclear why in Figure 7a the shape of the colour filled areas and the lines for the different scenarios not match (e.g. in July and August). I would recommend testing a visualisation as bar plot with a clear zero center value.
The colored areas did match the lines – it may have looked like they did not because we colored the area between zero and positive or negative values. We tried your suggestion to make the plot linear and included the result below. Initially, we chose the polar layout to emphasize that what happens in terms of storage in December, will affect the flood volumes in January. This is less clear in the linear layout. In addition, the polar figure was more attractive and original. However, we do agree that in the linear plot it is easier to read the values within one plot and to compare the values from the different plots. Therefore, we will include the new (linear) version in the revised manuscript.

[Figure]

**References**

Stoelzle, M., & Stein, L. (2021). Rainbow color map distorts and misleads research in hydrology–guidance for better visualizations and science communication. Hydrology and Earth System Sciences, 25(8), 4549-4565. Citation: https://doi.org/10.5194/egusphere-2025-1712-RC1

**1 Reviewer 2**

This study investigates how initial groundwater depth interacts with rainfall to influence pluvial flood volumes in lowland catchments under current and future climates. Using the WALRUS hydrological model and 109 years of hourly climate data, the authors simulate discharge and groundwater depth for 12 lowland catchments in and around the Netherlands. The work contributes to understanding compound flood drivers in flat, groundwater-dominated systems, which are often overlooked in large-scale or mountainous catchment studies.

Thank you for you positive opinion on the relevance of our study.

However, the findings about the groundwater-rainfall-flood relationship is not supervising. Additionally, the quantified findings have notable uncertainty, as they rely heavily on model outputs without thorough validation of the groundwater component.

Unfortunately, we were not entirely sure what you meant with "supervising". If you meant "surprising", we agree that it was not unexpected that we found a strong relation between initial groundwater depth, effective rainfall sum and discharge peaks. However, before this study we did not know this for certain and we did not

know exactly how strong the relation would be and what the exact effect of this relation would be on floods in a future climate.

In the response to the specific comments below, we explain how we will provide additional information about the calibration and validation procedure. We will also address the uncertainty in the model and effect of uncertainty on the robustness of the conclusions.

**General comments**

The study emphasizes the importance of shallow groundwater storage in controlling flood severity in lowlands. However, the groundwater-related variables are entirely estimated by the WALRUS model, which is calibrated only against streamflow data. This could introduce bias, especially when groundwater depth is the key factor for explaining flood generation mechanisms. How much confidence do the authors have in their estimates of groundwater depth?
This point has also been mentioned by Reviewer 1. Please see our detailed response there.

Moreover, the calibration period is short (e.g., one year), which is insufficient to capture the full range of hydrometeorological variability (e.g., droughts, wet years, seasonal shifts, extremes). This raises concerns about the generalizability of the model to future climates, which may introduce conditions absent in the calibration year. Please provide quantitative uncertainty assessments in the main text or supplementary materials.
Most catchments were indeed calibrated on one (some more) full year and validated on another year. For some catchments a longer period was used. More years for both calibration and validation would indeed be preferred to capture more of the interannual variability, but unfortunately longer time series of all necessary model input (in particular discharge and surface water supply) were not available for most catchments. In the revised version of the manuscript, we will explicitly mention the calibration period.

In a previous study (Brauer et al., HESS, 2014), we investigated the effect of different types of model uncertainty in detail. For this study, we used twelve catchments instead of one (or two) to represent the range of catchment circumstances which can be found in freely draining lowland areas. The intention of our study was not to project the exact changes for each catchment separately, but rather to give a range of possible directions and evaluate the dependency of flood frequency and severity on catchment characteristics. This allowed us, for example, to draw the conclusion that for the change in number of floods and flood volume, the differences between climate scenarios is larger than between catchments (Figure 9). We will make this intention more clear in the text of the manuscript.

The lowland catchment is a region closely related to human activities. How were these human impacts on groundwater depth and fluxes considered or approximated in the modeling framework?
Indeed the studied areas are affected by people. Groundwater extraction is limited – only in very dry summer months some water (surface water or deeper groundwater) is extracted for sprinkler irrigation. Surface water supply was included in the simulations, but since we do not know how this will change in the future, the amount and period of surface water supply was kept the same for all scenarios.

The study includes 12 catchments. Are their hydrogeological characteristics homogeneous? If not, how do these differences influence the groundwater–rainfall–flood interaction?

The hydrogeological characteristics differ between and within the catchments, though there are no impactful hydrogeological elements such as shallow bedrock, karstic fissures or large faults. The soils in all catchments consist of unconsolidated material of sedimentary origin, well beneath the groundwater table. (Loamy) sand is the major soil type of the top soil layer. There is naturally variation in topsoil material and aquifer thickness within each catchment. For example, the soils in the stream valleys often contain more loam. Since WALRUS is a lumped hydrological model, this heterogeneity is not resolved explicitly, but accounted for in a holostic manner. The simulated groundwater level therefore represents a catchment effective value which includes the variability. We will explain this in more detail in the revised version of the manuscript.

The WALRUS model includes two-way exchange between groundwater and surface water. Did this feedback mechanism significantly influence flood generation in the simulations? If so, please discuss and quantify where possible.
The groundwater-surface water interaction (flow between groundwater and surface water reservoir: fGS) impacts the runoff generation and discharge dynamics and thereby flood generation. It is not possible to turn this feature off in the model to investigate the effect of the surface water infiltration (or the effect which raised surface water levels have on limiting groundwater drainage) in isolation. In the revised version of the manuscript, we will include a figure in the supplementary material to investigate if there are correlations between the importance of the fGS flux on the one hand and flood frequency and flood severity on the other hand. We will plot the percentage increase or decrease in number of floods and flood volume presented in Figure 9 against the discharge dynamics metrics presented in Figure 3. For this, we will replace the baseflow index with the ratio between the sum of the flow from the groundwater reservoir to the surface water reservoir and the sum of the discharge, as suggested by both reviewers.

The WALRUS model includes bilateral flow exchange between groundwater and surface water. Did the flow direction between these water modules influence the flood generation in the simulations?
Please see our response to the previous comment.

**Specific comments**

L18-19: Typo
We will fix it.

L205: How much the difference between the estimated baseflow by Gustard and Demuth (2009) and the flow from the modelled groundwater?
Reviewer 1 asked the same thing and we investigated it. Please see our reply there.

Disucssion 4.2: The authors try to mix up the definition of soil wetness and groundwater in the discussion. It is conceptually in
This comment seems incomplete, but we assume that you meant to say the same as Reviewer 1. Thank you for pointing this out. We agree that it is important to avoid confusion. Regarding our own study, we used groundwater depth throughout the paper, while indeed emphasizing that there is a strong link between groundwater depth and topsoil wetness.

Figure 7: Better use full name of scenarios instead of acronym

The abbreviations are the formal abbreviations given to the scenarios by the Royal Netherlands Meteorological Institute. To avoid confusion, we kept those names. We do understand your confusion, so we added explanations to the captions of Table 2 and Figures 7, 8 and 9.

Figure S2: The range of values in the legend for Qpeak does not cover the full range of Qpeak values shown in the plot.

We will fix it in the revised manuscript.

---

## Author Response (AR2)

*Response to reviewer's comments after the re-submission of "Rain-on-wet-soil compound floods in lowlands: the combined effect of large rain events and shallow groundwater on discharge peaks in a changing climate" by Claudia Brauer, Ruben Imhoff and Remko Uijlenhoet, submitted to Hydrology and Earth System Sciences.*

We thank both reviewers for reviewing the text again. In the text below we respond to each item in green.

**Reviewer 1**

This study investigates how initial groundwater depth interacts with rainfall to influence pluvial flood volumes in lowland catchments under current and future climates by using the WALRUS hydrological model. The authors have substantially improved the quality of the manuscript. They now present more robust results after discussing the limitations and adding additional supplementary information in the revised version. Although all results are based on modelled data due to data unavailability, the findings are now more convincing. Therefore, I recommend that the manuscript be accepted for publication after a few minor corrections.

Specific comments:

L383: Should it be 2085 instead of 2050?
Thank you for pointing this out! We fixed it.

Figure S25 is cited earlier in the manuscript than Figures S9–S24. I suggest moving Figure S25 to an earlier position in the supplementary materials and updating the numbering accordingly.
We agree. We changed the order of the supplementary figures and references to those figures in the main text.

**Reviewer 2**

I would like to thank the authors for addressing most of my comments. I know only have some minor remarks which should be addressed.

The authors did include the article by Berghuijs & Slater (2023) as recommended, however it seems that the article was included only as perfunctory citation. Since it is one of the few articles that analyses floods and groundwater, it should be discussed more in depth. I am not a co-author on the mentioned article and genuinely only mention it since I see it as relevant for the discussion. How do their results compare to yours in regard to lowland catchments, and trends in groundwater contribution to flood?
We reread the paper and agree that there are similarities with our study that we should point out more clearly. We summarized the most important findings and added three sentences to the manuscript: in the Introduction section (L. 18 and L. 49 in the revised manuscript with tracked changes) and in the Discussion section (L. 457).

The comparison against groundwater table depth is very helpful. One more question about the modelling, does the timing of model calibration influence the representation of climate change in the model parameters? Some of the early and late calibration times are 15 years apart (e.g. two in 1998 (with validation in 1991-1992, Vechte A and Dinkel), several other more in the early or mid-2010 (Hupsel Brook, Grote Waterleiding, Radewijkerbeek).

The choice of calibration year indeed has an effect on the resulting parameter values, just as the calibration technique and person performing the calibration. We expect that the 'age' of the calibration dataset is less important than the particular weather during that year (wet or dry year). In addition, as we explained (L. 229 in the revised manuscript with tracked changes), "The intention of our study was not to project the exact changes for each catchment separately, but rather to give a range of possible directions for lowland catchments."

L186: what do you mean by "since time series of observations are too short for robust statistical analyses". What statistical analysis are you referring to?
With "statistical analyses" we meant all the analyses we did in this paper. The long time series allowed us to compute average changes and discharges belonging to certain return periods. With many years of data which include a large variety of weather conditions, the results and conclusions are less sensitive to the specific situation in a particular year, and this allowed us to draw conclusions about rare events. To make this more clear, we changed all three occurrences of this phrase in the final resived manuscript (L 90, 145 and 179).

Figure 5 (and its match in the supplement) is missing a colour scale legend.
There is indeed no legend, but the contour lines are labeled, allowing the reader to estimate the values. We added 'Colours represent the same values as the contour lines and are added to make the figures easier to interpret.' to the captions.

Supplement Figure S5: What do you mean by "Since the exact land elevation at the measurement locations was unknown, this had to be estimated." Dinoloket provides metadata for each station including elevation of the surface.
Thank you for pointing that out! We retrieved the metadata and made the validation plot again with the observed land surfaces. We deleted the sentence "Since ... estimated" in the caption.

**References**

Berghuijs, W. R., & Slater, L. J. (2023). Groundwater shapes North American river floods. Environmental Research Letters, 18(3), 034043.

---

## Author Response (AR3)

*Response to editor's comments after the re-submission of "Rain-on-wet-soil compound floods in lowlands: the combined effect of large rain events and shallow groundwater on discharge peaks in a changing climate" by Claudia Brauer, Ruben Imhoff and Remko Uijlenhoet, submitted to Hydrology and Earth System Sciences.*

**Editor**

Congratulations on your nice manuscript, which I am happy to provisionally accept for publication in HESS. Before final acceptance, I would like to encourage you to reconsider the color palettes used in Figures 1 and 5 (rainbow colors), which are not colorblind friendly.

Thank you for accepting the paper for publication!

We changed the color palettes of Figures 1 and 5, as well as Figure S8 in the supplement (showing Figure 5 for all catchments).